# Development of Quantitative Chemical Ionization Using Gas Chromatography/Mass Spectrometry and Gas Chromatography/Tandem Mass Spectrometry for Ambient Nitro- and Oxy-PAHs and Its Applications

**DOI:** 10.3390/molecules28020775

**Published:** 2023-01-12

**Authors:** Jungmin Jo, Ji-Yi Lee, Kyoung-Soon Jang, Atsushi Matsuki, Amgalan Natsagdorj, Yun-Gyong Ahn

**Affiliations:** 1Department of Environmental Science and Engineering, Ewha Womans University, Seoul 03760, Republic of Korea; 2Bio-Chemical Analysis Team, Korea Basic Science Institute, Cheongju 28119, Republic of Korea; 3Institute of Nature and Environmental Technology, Kanazawa University, Kanazawa 920-1192, Japan; 4Department of Chemistry, National University of Mongolia, Ulaanbaatar 14200, Mongolia; 5Western Seoul Center, Korea Basic Science Institute, Seoul 03759, Republic of Korea

**Keywords:** nitro-PAHs, oxy-PAHs, GC-NCI/MS, GC-PCI-MS/MS, aerosol samples

## Abstract

The concentration of polycyclic aromatic hydrocarbons (PAHs) in the atmosphere has been continually monitored since their toxicity became known, whereas nitro-PAHs (NPAHs) and oxy-PAHs (OPAHs), which are derivatives of PAHs by primary emissions or secondary formations in the atmosphere, have gained attention more recently. In this study, a method for the quantification of 18 NPAH and OPAH congeners in the atmosphere based on combined applications of gas chromatography coupled with chemical ionization mass spectrometry is presented. A high sensitivity and selectivity for the quantification of individual NPAH and OPAH congeners without sample preparations from the extract of aerosol samples were achieved using negative chemical ionization (NCI/MS) or positive chemical ionization tandem mass spectrometry (PCI-MS/MS). This analytical method was validated and applied to the aerosol samples collected from three regions in Northeast Asia—namely, Noto, Seoul, and Ulaanbaatar—from 15 December 2020 to 17 January 2021. The ranges of the method detection limits (MDLs) of the NPAHs and OPAHs for the analytical method were from 0.272 to 3.494 pg/m^3^ and 0.977 to 13.345 pg/m^3^, respectively. Among the three regions, Ulaanbaatar had the highest total mean concentration of NPAHs and OPAHs at 313.803 ± 176.349 ng/m^3^. The contribution of individual NPAHs and OPAHs in the total concentration differed according to the regional emission characteristics. As a result of the aerosol samples when the developed method was applied, the concentrations of NPAHs and OPAHs were quantified in the ranges of 0.016~3.659 ng/m^3^ and 0.002~201.704 ng/m^3^, respectively. It was concluded that the method could be utilized for the quantification of NPAHs and OPAHs over a wide concentration range.

## 1. Introduction

Atmospheric particulate matter (PM), which is composed of a significant proportion of carbonaceous compounds, is emitted directly into the atmosphere. A secondary organic aerosol is also formed from the photo-oxidation of mixed anthropogenic volatile organic compounds [1]. Among the organic compounds related to PM, polycyclic aromatic hydrocarbons (PAHs) are the representative air toxic substances that are regulated because of their carcinogenic and mutagenic properties [2,3,4]. PAHs mainly originate from the incomplete combustion of coal, vehicle exhausts, and biomass burning; nitro-PAHs (NPAHs) and oxy-PAHs (OPAHs) can also arise by a secondary formation from the reactions of PAHs with atmospheric oxidants [5,6]. They are known to be much more toxic than the parent PAHs. Accordingly, increasing attention has been paid to research topics such as their formation mechanism, toxicity, source identification, and risk assessments [7,8,9]. For example, the value of the toxic equivalent factor for 6-nitrochrysene, which is an NPAH congener, is 10 times higher than that of Benzo[a]pyrene in the PAH group [10]. Thus, although the concentrations of NPAHs and OPAHs are generally lower than those of their parent PAHs in the atmosphere, increasing attention has been paid to these PAH derivatives. Furthermore, their monitoring in the atmosphere, as well as that of PAHs, is becoming more important in order to determine whether their formation is due to primary emissions or secondary sources [11,12,13,14,15]. However, quantitative data of PAH derivatives in aerosol samples are much fewer than those for parent PAHs because it is difficult to quantify them using the universal method applied to parent PAHs. The concentrations of PAH derivatives, especially NPAHs in aerosol samples, have been detected at 10~100 times lower than those of parent PAHs [16,17,18]. Thus, a sensitive analytical method with a high selectivity that considers their trace concentration level is required [19].

Several analytical methods have been employed for the determination of PAH derivatives [20]. In the case of NPAH measurements, gas chromatography (GC) combined with various detection methods such as electron capture detection [21,22], nitrogen selective detection [23,24], reductive electro-chemical detection [25,26], and negative and positive chemical ionization (NCI and PCI) mass spectrometry (MS) have been reported [27,28,29,30]. In the method of electron ionization mass spectrometry, the study of PAHs and nitro-PAHs using thermal desorption gas chromatography has been reported [31]. Among these methods, GC coupled with MS has been the most used [32,33]. Furthermore, the application of tandem mass spectrometry (MS/MS) by using a triple quadrupole mass spectrometer (QqQ-MS) has recently increased in terms of analytical sensitivity and specificity enhancement [34,35]. There are two common ionization methods, electron impact (EI) and chemical ionization (CI), in a GC/MS analysis. The EI method is commonly used to quantify parent PAHs because of its advantages such as a mass spectral library search to assist compound identification by providing a reference for mass spectra [36,37]. In the case of the CI method, negative chemical ionization (NCI) and positive chemical ionization (PCI) can be selected according to the chemical properties of the target analytes. As NPAHs and OPAHs exist at lower concentration levels in the atmosphere than parent PAHs, the CI method has been attempted more often than EI for a sensitive and selective detection [38]. Consequently, a rapid and accurate quantitative analytical method is required; this method, without a sample preparation, is preferred due to the difficulty in procuring samples and having to obtain a variety of chemical information from one specific sample alone. Nicol et al. developed a GC-MS/MS analysis of the EI method for OPAHs in an air concentration range of 30–170 pg/m^3^ without a purification of the extract [39]. The detection limits of NPAHs and OPAHs reported so far are mostly results based on the signal-to-noise ratio; the method detection limit (MDL) has not yet been evaluated. In view of this, the aims of this study were: (1) to optimize the analytical conditions for 8 NPAH and 10 OPAH congeners using negative chemical ionization mass spectrometry (NCI/MS) or positive chemical ionization tandem mass spectrometry (PCI-MS/MS); (2) to validate the method with respect to the linearity, recovery, and MDL; and (3) to assess the practical applicability for the quantification of NPAHs and OPAHs over a wide concentration range in aerosol samples collected from Northeast Asian sites.

## 2. Results and Discussion

### 2.1. Congener-Specific Determination of PAH Derivatives

#### 2.1.1. Optimal Ionization Mode Selection

The use of NCI can be the most selective detection method for specific classes of molecules containing electro-negative or acidic groups [40]. The mass spectra of 1-nitronaphthalene (1-NNAP) obtained by each ionization mode is shown in Figure 1. The most intense peak of the molecular ion for 1-NNAP is shown in the two types of chemical ionization modes. In the EI mode, the mass spectrum (Figure 1a) contained many fragmented ions and less sensitive molecular ions. In the PCI mode (Figure 1c), where the protonated molecular ion [M+H]^+^ was generated by methane gas, added ions such as [M+C_2_H_5_]^+^ and [M+C_3_H_5_]^+^ as well as a fragmented ion [M-CH_3_]^+^ from the loss of a methyl group were identified. On the other hand, the strongest molecular ion, [M-H]^-^, was produced in the NCI mode (Figure 1b). As a result, the NCI mode provided a high sensitivity and selectivity for the detection of 1-NNAP. When the intensities of the NPAH molecular ions were compared with the NCI and EI modes, their sensitivities were 3 to 15 times higher under the NCI condition (Appendix A).

#### 2.1.2. Ionization Efficiency

In a GC-MS/MS analysis, the molecular ions of the target analytes in the two types of chemical ionization mode are selected as the precursor ion that is then fragmented by a collision-induced dissociation (CID) in MS/MS to generate the product ions [41]. Despite the advantage of GC-MS/MS in that it is capable of a high-sensitivity analysis, low ionization efficiencies of the product ions from NPAH molecular ions were obtained because of the formation of strong and stable molecular ions in the case of the NCI condition. Consequently, the data acquisitions for the analysis of atmospheric NPAHs and OPAHs were performed by selected ion monitoring (SIM) under the NCI condition and selected reaction monitoring (SRM) under the PCI condition. Most NPAHs showed a better sensitivity in the SIM mode under the NCI condition, but the use of the PCI mode under the SRM condition was more appropriate in terms of the sensitivity and selectivity for the OPAHs, as shown in Figure 2. The sensitivities of xanthone (XT), phenalenone (PH), and 5,12-naphthacenequinone (Ncq) in the class of OPAHs and 6-nitrochrysene (6-NCHR) in the class of NPAHs particularly showed a significant difference between the two detection methods.

The results obtained by GC-NCI/MS and GC-PCI-MS/MS in comparing the sensitivities of four analytes at three different concentrations (Figure 3) showed that the detection method for individual OPAHs and NPAHs was determined through results that provided a clear change of response sensitivity in accordance with the different concentrations.

#### 2.1.3. Chromatographic Separation

For the quantification of the target PAH derivatives from multiple components in the extracts of actual samples, the separation between the individual PAH derivatives and the neighborhood interferences obtained by GC-NCI/MS and GC-PCI-MS/MS were investigated. Even though NCI has the advantage of a high selectivity for NPAHs, 9-nitroanthracene (9-NANT) was not completely separated from nearby interferences in the actual aerosol samples when GC-NCI/MS was used. In contrast, a complete separation was achieved when GC-PCI-MS/MS was used (Figure 4b). GC-NCI/MS was superior for most NPAH congeners except for 9-NANT in terms of the sensitivity and selectivity. A typical case where the peak of 2-nitrofluorene (2-NFLUO) was only confirmed when GC-NCI/MS was used at a concentration of 0.034 µg/m^3^ is shown in Figure 4a. In the case of the OPAH congeners, GC-PCI-MS/MS could offer a higher sensitivity and selectivity than GC-NCI/MS such as for the PH shown in Figure 4c through the use of the SRM mode.

### 2.2. Method Validation

The optimized conditions for the quantification of individual NPAH and OPAH congeners through a consideration for the sensitivity and selectivity in actual samples is shown in Table 1. For an internal calibration, fluoranthene-d_10_ (Fla-d_10_), which has a high efficiency for ionization and separation, was selected among the label standards used in the analysis of PAHs.

The range of the calibration curve was determined within a quantifiable range of individual NPAH and OPAH congeners in actual samples. The concentration of NPAHs was at least 10 times lower than that of OPAHs in the actual samples and this difference of concentration was reflected in the spiking experiments for the method validation. The calibration curves were linear in the range of each congener presented in Table 2 and their correlation coefficients were all greater than 0.99. The MDLs were measured by analyzing seven replicate samples spiked with 0.01–0.06 ng of the NPAHs and OPAHs per a control sample free of the target analytes. MDL was defined as the lowest concentration of each NPAH and OPAH congener that provided a greater than 99% confidence when the analytical method was used [42]. The recovery experiment was evaluated with two expected concentrations in the actual aerosol samples. MDLs for the NPAH and OPAH congeners ranged from 0.272 to 3.494 pg/m^3^ and 0.977 to 13.345 pg/m^3^, respectively.

### 2.3. Application to Actual Aerosol Samples

For the aerosol samples collected in Noto, Seoul, and Mongolia, an analysis was only carried out when the correlation coefficients of the calibration curves for the NPAH and OPAH congeners were above 0.99. Quality control samples prepared at the middle concentration of the calibration curve were analyzed together every time after each 15-sample analysis. The QC results used to validate the analytical results obtained for every batch during the period of the sample analysis are shown in Figure 5.

A re-analysis was performed when the result value was outside the ± 2SD (standard deviation) limit through the QC result confirmation for individual NPAH and OPAH congeners obtained from their specific detection method. Most NPAH and OPAH congeners did not go out of the ±2SD limit, except for 9-NANT. When analyzing high concentration levels (such as the samples collected in Mongolia), the QC result was out of the acceptable limit. It was found that a replacement of the GC liner or the cleaning of the ion source were required for these times. From the results of measuring the NPAHs and OPAHs in actual aerosol samples (*n* = 84) using a validated analytical method, the total mean concentrations (NPAHs + OPAHs) in the three regions were 0.009 ± 1.009 ng/m^3^ for Noto, 15.394 ± 3.134 ng/m^3^ for Seoul, and 313.803 ± 176.349 ng/m^3^ for Ulaanbaatar, respectively. The difference between the mean concentrations of the NPAHs and OPAHs was about 20–530-fold, depending on the region.

The ranges of the minimum and maximum concentrations of the NPAHs and OPAHs in all regions where the concentration was above the MDL are shown in Figure 6. The concentration range, mean value (ng/m^3^), and detection frequency of each NPAH and OPAH congener are presented in Appendix A.

The measured concentrations of individual NPAH and OPAH congeners ranged from 0.002 to 201.704 ng/m^3^. The ranges of the measured concentration in the aerosol samples using GC-NCI/MS and GC-PCI-MS/MS in this study were 0.016–3.659 ng/m^3^ and 0.002–201.704 ng/m^3^, respectively. The detection rate of the number of target analytes (18 NPAH and OPAH congeners) whose concentration was calculated to be higher than the MDL in the actual samples based on the total number of analytes per sample was found to be 27% in the samples from Noto compared with 90% from Ulaanbaatar and 94% from Seoul. When OPAHs with a high detection rate in the air were compared from the two urban areas of Ulaanbaatar and Seoul (except for Noto, where 73% was not detected), the difference in the maximum average concentration was 55 times. Among the detected 10 OPAH congeners in the samples from Seoul and Ulaanbaatar, the major congeners were 9-fluorenone (9-Flu), anthraquinone (Anq), 1,8-naphthalic anhydride (1,8-NA), and PH. Among these four congeners, the concentrations ranged from 16.703–201.704 ng/m^3^ for 9-Flu and 12.549–174.388 ng/m^3^ for PH. From the results for the NPAHs, the maximum concentration in the sample from Seoul was 0.387 ng/m^3^ for 9-NANT. The analytical results of the samples from Ulaanbaatar showed that 3-nitrofluoranthene (3-NFL) had the highest concentration of 3.659 ng/m^3^. 3-NFL, 9-NTNT, and 2-nitronaphthalene (2-NNAP) were the main congeners detected in the samples. 9-NANT was the main congener detected in the samples from both Seoul and Ulaanbaatar. It has been reported that 9-NANT is emitted from direct sources, predominantly diesel engines, and is also formed in the atmosphere as a result of secondary reactions [43,44]. Ulaanbaatar generally uses biomass fuel and coal fuel (lignite) for household cooking and heating It is an urban area with high levels of PM and PAH pollution because of the use of solid fuels [45]. In Ulaanbaatar, the concentration of PAHs is high, and this causes high concentrations of NPAHs and OPAHs because of secondary formations. Additionally, the main causes for the primary emissions of NPAHs in urban regions are diesel engine vehicles and incomplete combustion by residential heating [44]. As biomass combustion is the main source of OPAHs [46], biomass and heating fuel-use in Ulaanbaatar could contribute to high NPAH and OPAH concentrations. The contribution of individual NPAH and OPAH congeners to the regional total concentration was different because of the different characteristics of regional occurrence. The proposed analytical method for the monitoring of aerosol samples could be of practical use to reveal such a source of PAH derivatives.

## 3. Materials and Methods

### 3.1. Sampling Sites and Method

The aerosol samples were collected at three sites in winter from 15 December 2020 to 17 January 2021. Two sites were the capital cities of South Korea (Seoul) and Mongolia (Ulaanbaatar), and the other site was a background region in Japan (Noto). The sampling locations are shown in Appendix A. The sampling time was 24 h, from 10:00 a.m. to 9:00 p.m. the next day, and a high-volume air sampler (Shibata, Tokyo, Japan, HV-1000RW) equipped with an impactor for PM2.5 was used. Quartz fiber filters (QFFs, Tissuequartz 2500QAT-UP, 8 × 10 in, Pall, USA) were baked at 550 °C for 12 h before sampling to remove the organic matter. All sampled filters were wrapped in pre-baked aluminum foil and stored at −20 °C until the analysis.

### 3.2. Sample Extraction and Analysis

The analytes collected on the QFFs were extracted twice for 30 min each with a mixture of dichloromethane and methanol (DCM:MeOH at a ratio of 3:1 *v/v*) by sonication. Before the extraction, Flu-d10 was added as an internal standard and spiked at 780 ng. The extract was filtered through a 0.45 μm nylon membrane filter and concentrated to 1 mL using TurboVap (TurboVap LV, Zymark, Germany). After being transferred to a vial, nitrogen was concentrated to 0.5 mL at 35 °C or less and MS was performed. The QFF field blanks were analyzed for each field sample and processed using the same procedure.

All standard reagents for the analytes were purchased from AccuStandard Chemical (USA); the target analytes are listed in Appendix A. The NPAHs and OPAHs were analyzed by gas chromatography (7890B GC) coupled with 5977A MS (Agilent Technologies, USA) and 7010 triple quadrupole MS (Agilent Technologies, USA) with interchangeable EI and CI ion sources. Chemical ionization was used for the target substances of 8 NPAHs and 10 OPAHs. The quantitative conditions for NCI and PCI were optimized using standard reagents for each analyte. NCI and PCI were quantified by SIM and SRM methods, respectively. The analytes were classified by a DB-5MS UI column and analyzed in NCI-GC/MS and PCI-GC-MS/MS modes. High-purity helium (99.999%) was used as the carrier gas, with a flow rate of 1 mL/min. The reagent gas was high-purity methane (99.999%) and the flow rate was 1 mL/min for the NCI mode and 2 mL/min for the PCI mode. All samples were injected with 2 μL in a splitless mode at 300 °C. The GC oven conditions were as follows: 60 °C → 145 °C (10 °C/min) → 220 °C (4 °C/min) → 320 °C (10 °C/min, hold for 20 min). Detailed information on the analysis conditions are summarized in Appendix A.

### 3.3. Quality Assurance and Quality Control

Linearity, MDL, and recovery experiments of the calibration curve were performed to evaluate the QA/QC for the assay method. For the MDL, the lowest concentration was added to the filter and the experiment was repeated 7 times in the same way; the MDL was obtained from the standard deviation of the measured concentration. Thus, the procedure was carried out through three repeated experiments. In addition, in order to reduce analytical errors that may have occurred in the continuous analytical measurements and to accurately measure the results, the QC substances were periodically measured for every 15 sample batches. The analysis was performed whilst taking necessary precautions so that the accuracy of the measured value and the expected value of the QC material for which the exact value was known did not deviate from the allowable limit. QA/QC was validated for each analyte according to the optimized NCI and PCI analytical methods.

## 4. Conclusions

Congener-specific methods were evaluated for the quantification of NPAHs and OPAHs, which are PAH derivatives. For the NPAHs, GC-NCI/MS generated strong molecular ions and showed a better sensitivity than GC-PCI-MS/MS. However, for 9-NANT, it was difficult to obtain a complete separation from nearby interferences in the actual aerosol samples; thus, GC-PCI-MS/MS was used. In the case of all OPAHs, the background was reduced in the sample matrix by means of only the product ion being quantified from the specific parent ion for an individual OPAH congener using GC-PCI-MS/MS with the SRM mode. This enabled a highly sensitive and selective quantitative analysis for the OPAHs. For the method validation, the linearity of the calibration curve, MDL, and recovery were evaluated for each individual congener. In the actual aerosol samples collected at Noto, Seoul, and Ulaanbaatar, the analytical stability of each batch was confirmed from the results of the QC samples. A re-analysis was performed if the result was out of the allowable range. Among the three regions, the total mean concentrations of NAPHs and OPAHs were highest in Ulaanbaatar, followed, in order, by Seoul and Noto. The most abundant NPAHs in Seoul and Ulaanbaatar were 9-NANT and 3-NFL; among the OPAHs, there were high concentrations of 1,8-NA and 9-Flu. With the analysis method in this study, the concentrations of NPAHs and OPAHs in the samples of atmospheric PM2.5 could be quantified in the range of 0.016 to 3.659 ng/m^3^ and 0.002 to 201.704 ng/m^3^, respectively. This method allowed the quantification of NPAHs and OPAHs over a wide concentration range by region, and could be of help in the field of aerosol research.

## Figures and Tables

**Figure 1 molecules-28-00775-f001:**
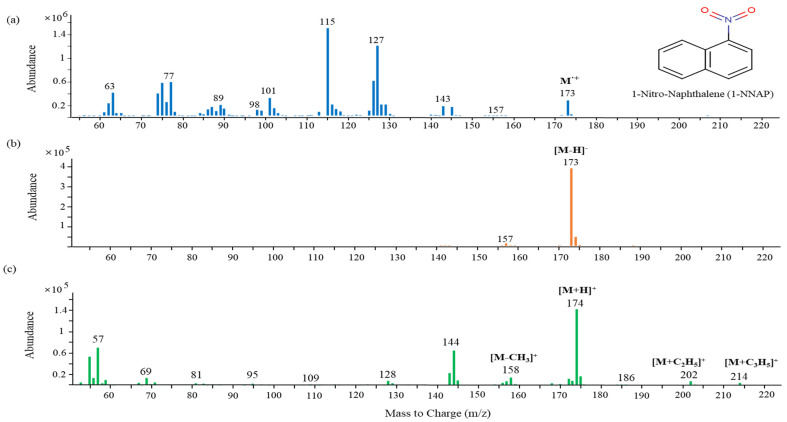
Mass spectra of 1−nitronaphthalene obtained from each ionization mode: (**a**) EI; (**b**) NCI; (**c**) PCI.

**Figure 2 molecules-28-00775-f002:**
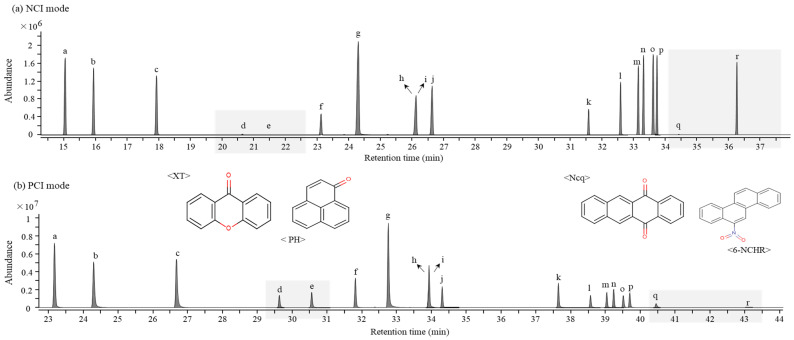
Comparison between SIM and SRM modes of a standard solution with NPAHs and OPAHs (4 μg/mL): (**a**) SIM chromatogram under NCI condition on DB-5MS UI capillary column (30 m × 0.25 mm × 0.25 μm); (**b**) SRM chromatogram under PCI condition on DB-5MS UI capillary column (60 m × 0.25 mm × 0.25 μm). The gray shading indicates four congeners, with a significant difference in sensitivity between the two detection methods. The peak identities were as follows: a—1-nitronaphthalene (1-NNAP); b—2-nitronaphthalene (2-NNAP); c—9-fluorenone (9-Flu); d—xanthone (XT); e—phenalenone (PH); f—anthraquinone (Anq); g—1,8-naphthalic anhydride (1,8-NA); h—2-nitrofluorene (2-NFLUO); i—2-methylanthraquinone (2-Maq); j—9-nitroanthracene (9-NANT); k—benzo[b]fluoren-11-one (BbFLU); l—menzoanthrone (BZA); m—3-nitrofluoranthene (3-NFL); n—4-nitropyrene (4-NPYR); o—benz[a]anthracene-1,2-quinone (BAQ); p—1-nitropyrene (1-NPYR); q—5,12-naphthacenequinone (Ncq); r—6-nitrochrysene (6-NCHR).

**Figure 3 molecules-28-00775-f003:**
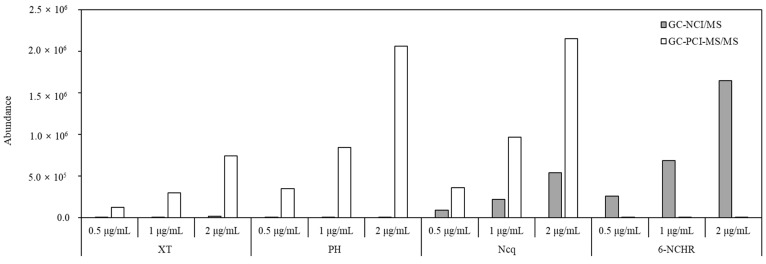
Comparison of the sensitivity of XT, PH, Ncq, and 6-NCHR at three different concentrations obtained by GC-NCI/MS and GC-PCI-MS/MS.

**Figure 4 molecules-28-00775-f004:**
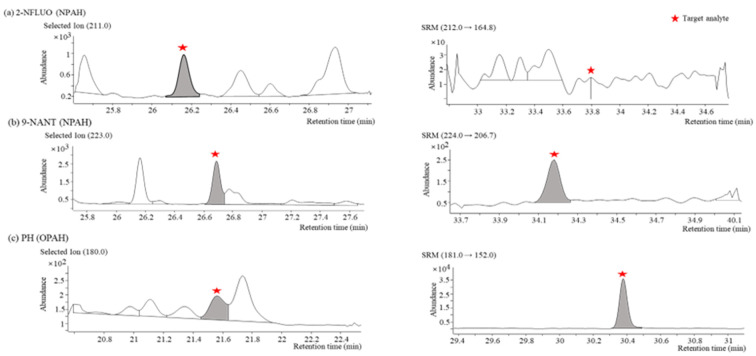
Comparison of separation characteristics of target PAH derivatives in actual aerosol samples obtained by GC-NCI/MS (**left**) and GC-PCI-MS/MS (**right**): (**a**) SIM (*m/z* = 211) and SRM (*m/z* 212→164.8) chromatograms of 2−NFLUO (NPAH); (**b**) SIM (*m/z* = 223) and SRM (*m/z* 224→206.7) chromatograms of 9−NANT (NPAH); (**c**) SIM (*m/z* = 180) and SRM (*m/z* 181→152) chromatograms of PH (OPAH).

**Figure 5 molecules-28-00775-f005:**
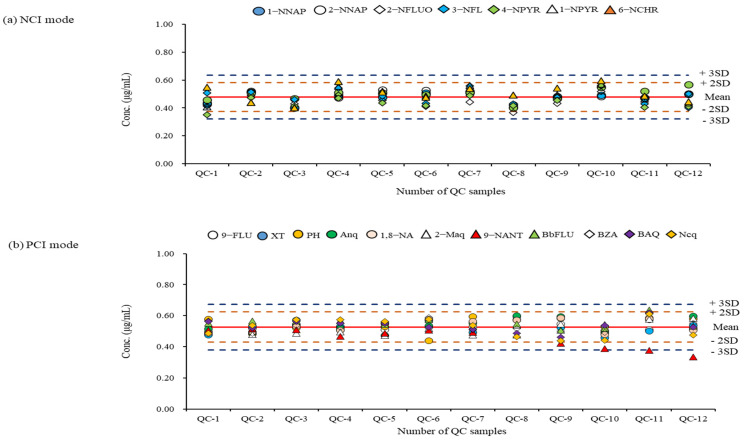
QC results during the period of sample analysis: (**a**) QC chart of analytes measured in NCI mode; (**b**) ) QC chart of analytes measured in PCI mode. The graph indicates that upper and lower warning and rejection limits for reference standard materials (0.5 ug/mL).

**Figure 6 molecules-28-00775-f006:**
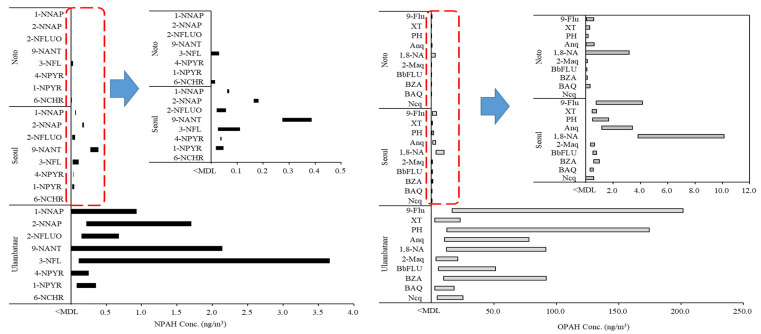
Concentration ranges of individual NPAH and OPAH congeners in PM_2.5_ from real samples.

**Table 1 molecules-28-00775-t001:** Selected ions for the quantification of individual NPAH and OPAH congeners in NCI-SIM mode and transitions in PCI-SRM mode for MS detection.

NCI Conditions			PCI Conditions		
Compound	Qualifier Ion (*m/z*)	Quantifier Ion (*m/z*)	Compound	SRM (*m/z*)	Collision Energies (eV)
1-NNAP	157	173	9-Flu	181.0 → 152.0	30
2-NNAP	157	173	XT	197.0 → 115.1	50
Fla-d10 (IS)	213	212	PH	181.0 → 152.0	30
2-NFLUO	188	211	Anq	209.0 → 151.9	30
3-NFL	231	247	1,8-NA	199.0 → 115.0	30
4-NPYR	231	247	Fla-d10	213.0 → 182.8	30
1-NPYR	231	247	2-Maq	223.0 → 152.0	30
6-NCHR	257	273	9-NANT	224.0 → 206.7	5
			BbFLU	231.0 → 202.0	30
			BZA	231.0 → 202.0	30
			BAQ	259.0 → 202.0	50
			Ncq	259.0 → 202.1	50

**Table 2 molecules-28-00775-t002:** Calibration linear regression value, MDLs, and recoveries of individual NPAH and OPAH congeners obtained by the analytical method.

NPAHs	Calibration		MDL(pg/m^3^) ^a^	Recovery ± RSD% (*n* = 3)	
	Range (μg/mL)	R^2^		0.02 ng/m^3^	0.04 ng/m^3^
1-NNAP	0.01–0.4	0.997	0.551	103.8 ± 11.3	104.8 ± 4.0
2-NNAP	0.01–0.4	0.997	0.527	111.7 ± 13.3	102.3 ± 9.6
2-NFLUO	0.01–0.2	0.996	0.339	91.4 ± 3.5	116.5 ± 0.7
9-NANT	0.05–1	0.995	3.494	94.6 ± 1.6	98.3 ± 1.2
3-NFL	0.01–0.1	0.995	0.272	104.3 ± 10.2	102.0 ± 2.1
4-NPYR	0.01–0.2	0.997	0.494	94.8 ± 9.1	92.7 ± 2.9
1-NPYR	0.01–0.1	0.999	0.494	92.8 ± 12.6	103.9 ± 2.3
6-NCHR	0.01–0.2	0.997	0.298	110.0 ± 8.6	105.7 ± 2.6
**OPAHs**	**Calibration**		**MDL(pg/m^3^)**	**Recovery ± RSD% (*n* = 3)**	
	**Range (** **μ** **g/mL)**	**R^2^**		**0.2 ng/m^3^**	**0.9 ng/m^3^**
9-Flu	0.1–5	0.996	0.977	104.0 ± 3.9	92.8 ± 11.6
XT	0.2–2	0.998	5.731	107.9 ± 1.3	100.1 ± 7.3
PH	0.1–4	0.998	6.216	98.2 ± 2.3	98.6 ± 5.3
Anq	0.1–4	0.994	3.128	106.8 ± 2.1	110.0 ± 6.5
1,8-NA	0.1–5	0.999	0.873	103.3 ± 1.5	102.9 ± 2.2
2-Maq	0.1–2	0.995	2.341	104.4 ± 1.0	89.1 ± 2.0
BbFLU	0.2–4	0.994	2.789	108.6 ± 2.3	106.9 ± 1.1
BZA	0.2–4	0.996	4.666	107.7 ± 0.2	101.7 ± 2.1
BAQ	0.2–4	0.995	4.328	104.6 ± 3.4	108.8 ± 1.4
Ncq	0.2–4	0.998	13.345	102.9 ± 1.9	101.7 ± 2.4

^a^ MDL = t(*n* − 1, 1 − α = 0.99) X SD, t(6, 0.99) = 3.14 (*n* = 7).

## Data Availability

The data presented in this study are available on request from the corresponding author.

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
