# Peer review of "Development of Quantitative Chemical Ionization Using Gas Chromatography/Mass Spectrometry and Gas Chromatography/Tandem Mass Spectrometry for Ambient Nitro- and Oxy-PAHs and Its Applications"

_molecules, 2023, doi:10.3390/molecules28020775_

Round 1

Reviewer 1 Report

The authors report chromatographic-mass spectrometry method development for analysis of oxy-PAHs. Although, these methods are available in the literature analysis of congeners always  present challenges. The existence of this data in Asia is very limited. Does the work presented by authors is acceptable.

However, they are minor issues need to be addressed:

1. Authors need to list chemicals and instrument where there were bought in the method section for easy repeatation of the proposed method.

2. fix line 47 (grammar)

3. line 49 insert a comma before which

4. line 100 sentence is too long

5. move caption for figure 6 to the same page as the figure

Author Response

Dear Reviewer,

Thank you for  taking the time to review our work.

We have adopted most of the comments and suggestions and have revised the paper accordingly. The changes are marked in red in the paper.

Comment 1. Authors need to list chemicals and instrument where there were bought in the method section for easy repetition of the proposed method.

Response:

A list of chemicals was provided in Table S3. Also, their provider and instrument manufacturer were described in Section 3.2 (3.2. Sample extraction and analysis). A summary of instruments and analytical conditions was summarized again in Table S3. Please check the file of Supplementary.

Comment 2 & 3. fix line 47 (grammar) & line 49 insert a comma before which

Response:

We corrected and changed the parts as you pointed out. (page 2, line 48-51)

Comment 4.  line 100 sentence is too long

Response:

We modified the long sentence by dividing it into two sentences. 

(page 3, line 112-114)

Comment 5. move caption for figure 6 to the same page as the figure

Response:

The caption of Figure 6 was moved to the same page as the corresponding figure as you pointed out. (page 8)

Reviewer 2 Report

Authors propose an interesting analytical strategy in order to evaluate the presence of polycyclic aromatic hydrocarbons (PAHs) in particle matter (PM). Particular emphasis was paid in the development of a sensitive and selective analytical method for the detection of nitrogen and oxygen-based PAHs by using tandem (MS/MS) mass spectrometer. The interest around the PAH in particular matter is significantly increasing in the laboratories involved in environmental analysis, consequently the development of novel and innovative strategy is well-liked. The idea to optimize two different acquisition methods for the acquisition of target analytes in only one chromatographic run is very interesting. Very low detection limits were reached, showing that the proposed strategy was very sensitive.

The manuscript contains several sections and/or issues that should be revisited by the Authors. For example, the conclusion that the chemical ionization (in negative and positive mode) is better than to electron ionization (EI) in term of molecular ion intensity, should be supported by experimental values. In order to affirm these results, the Authors should consider the absolute intensity of each fragment ion. Also, please insert a table containing the absolute values of the molecular ions obtained using different ionization modes.

In my opinion, the submitted article can be accepted for the publication in MOLECULES, but after major revision. The following suggestions should be considered:

The title of the manuscript should be modified considering that the detection of the analytes was carried out by using tandem mass spectrometry technique.

Introduction Section, line 59: “quantitative data is much less than that for PAHs.” This sentence is not clear. Please modify the phrase

Introduction Section, lines 61-64: Why is not described the electron ionization? Please, comment it.

Introduction Section, lines 61-64: Why is not described the electron ionization? Please, comment it.

Introduction Section, line 66: “tandem mass spectrometry”. Please, report the acronym of tandem mass spectrometry (MS/MS)

Introduction Section, line 66: Please, report “by using triple quadrupole instrument (QqQ-MS)”

Introduction Section, line 67: “of quantitative analysis in various matrices”. The use of the triple quadrupole allows to improve the sensitivity and sensibility of the analysis. Please, change the sentence.

Introduction Section, lines 67-78: In opinion of this Reviewer, the section on the use of mass spectrometry should be reformulated in an orderly manner because in its current form it is quite confusing.

Results and Discussion Section, Figure 1: In opinion of this Reviewer, the figure 1 should be modified. Please, report the same m/z intensity (e.g., 190 m/z) for all MS spectra.

Results and Discussion Section, line 113: “Despite 111 the advantage of GC-MS/MS that it is capable of high-sensitivity analysis, it did not pro-112 vide sufficient ionization efficiency of the product ion because of the formation of strong 113 and stable molecular ions of the NPAHs under the NCI condition.” "In opinion of this Reviewer, this sentence should be modified according to low ionization efficiency obtained in ONLY negative chemical ionization condition.

Results and Discussion Section, Figure 2: In opinion of this Reviewer, the retention time (min) scale should be of equal intensity for a direct comparison of the chromatograms.

Results and Discussion Section, Figure 2 Caption: The Authors should report the extend name of target analytes. This is due to the fact that the acronyms have not yet reported in the text.

Results and Discussion Section, Table 1: Please, change the Confirmation ion and Quantification ion in Qualifier ion and Quantifier ion, respectively.

Materials and Methods Section: Please, report the mass spectrometry used for the detection of the analytes.
